# Particulate Matter Exacerbates the Death of Dopaminergic Neurons in Parkinson’s Disease through an Inflammatory Response

**DOI:** 10.3390/ijms23126487

**Published:** 2022-06-10

**Authors:** Dabin Choi, Gaheon Lee, Kyung Hwa Kim, Hyunsu Bae

**Affiliations:** 1Department of Physiology, College of Korean Medicine, Kyung Hee University, 26-6 Kyungheedae-ro, Dongdaemoon-gu, Seoul 02453, Korea; dphs0228@naver.com; 2Department of Health Sciences, The Graduate School of Dong-A University, 840 Hadan-dong, Saha-gu, Busan 49315, Korea; jingun1984@naver.com

**Keywords:** Parkinson’s disease, particulate matter, neuroinflammation, systemic inflammation, microglial activation

## Abstract

Particulate matter (PM), a component of air pollution, has been epidemiologically associated with a variety of diseases. Recent reports reveal that PM has detrimental effects on the brain. In this study, we aimed to investigate the biological effects of ambient particles on the neurodegenerative disease Parkinson’s disease (PD). We exposed mice to coarse particles (PM_10_: 2.5–10 μm) for short (5 days) and long (8 weeks) durations via intratracheal instillation. Long-term PM_10_ exposure exacerbated motor impairment and dopaminergic neuron death in 1-methyl-4-phenyl-1,2,3,6-tetrahydropyridine (MPTP)-induced PD mouse models. Short-term PM_10_ exposure resulted in both pulmonary and systemic inflammatory responses in mice. We further investigated the mechanism underlying PM_10_-induced neurotoxicity in cocultures of lung LA-4 epithelial cells and RAW264.7 macrophages. PM_10_ treatment elicited a dramatic increase in proinflammatory mediators in LA-4/RAW264.7 coculture. Treating BV2 microglial cells with PM_10_-treated conditioned medium induced microglial activation. Furthermore, 1-methyl-4-phenylpyridinium (MPP^+^) treatment caused notable cell death in N2A neurons cocultured with activated BV2 cells in PM_10_-conditioned medium. Altogether, our results demonstrated that PM_10_ plays a role in the neurodegeneration associated with PD. Thus, the impact of PM_10_ on neurodegeneration could be related to detrimental air pollution-induced systemic effects on the brain.

## 1. Introduction

Air pollution is a well-known risk factor for respiratory and cardiovascular diseases [1]. Interestingly, increasing evidence has suggested that air pollution is associated with neurological disorders, including stroke, Alzheimer’s disease (AD), and Parkinson’s disease (PD) [2,3]. Among air pollution components, particulate matter (PM) is a well-documented detrimental component that affects brain health [4]. Notably, long-term exposure to traffic-related air pollution over 20 years increased cognitive impairments in elderly people, which was correlated with the level of coarse particles with diameters between 2.5 and 10 µm (PM_10_) in the urbanized area [5]. Even in healthy children and young adults, long-term exposure to high concentrations of ambient PM results in impaired cognition and enhanced neuroinflammation [6]. Over the last two decades, this noticeable association between PM and brain disorders has gradually been confirmed and extended to humans and animal models. However, the precise mechanisms underlying this association remain unclear.

Parkinson’s disease is a chronic disorder characterized by the progressive and selective loss of dopaminergic (DA) neurons [7]. While the etiology of PD remains unknown, a considerable body of research has suggested that the neuroinflammatory response plays undeniable key roles in the loss of DA neurons during PD progression [8]. Specifically, accumulating evidence has revealed enhanced microglial activation and inflammatory processes in the brain during PD progression [9], suggesting that microglia play a significant role in the neuroinflammation associated with the pathogenesis of PD.

Indeed, microglia are resident immune cells in the CNS that actively survey the environment in the brain [10]. Unlike PM10, fine PM (PM_2.5_ < 2.5 μm) is able to penetrate deeply into the lungs [11]. Recent epidemiologic studies have reported that long-term exposure to PM_2.5_ can accelerate the development of PD and AD [12]. The treatment of microglial BV2 cells with PM_2.5_ elicited the upregulation of proinflammatory cytokines, including IL-1β and TNF-α [13]. When proinflammatory cytokines and chemokines are released from the blood to the brain via the blood–brain barrier (BBB), microglia are activated, and the activated microglia subsequently result in disastrous neurotoxic consequences, such as neuroinflammation and neurodegeneration [14]. However, the impact of PM_10_ on neurodegeneration remains unclear.

Taking these observations into consideration, we hypothesized that PM_10_ may have a detrimental effect on neurodegeneration in PD through its effects on systemic inflammation. To test this hypothesis, we investigated the neurotoxic and inflammation-modulating effects of PM_10_ on 1-methyl-4-phenyl-1,2,3,6-tetrahydropyridine (MPTP)-induced PD mouse models. To investigate the neurotoxic effects of PM_10_ on chronic PD, we exposed mice to PM_10_ for 8 weeks and then treated them with MPTP to generate an animal model of PD. The inflammation-modulating effects of PM_10_ on mice were further determined by short-term exposure to PM_10_ for 5 days. Additionally, the mechanisms underlying PM_10_-induced neurotoxicity were analyzed with in vitro experiments with cocultures of lung LA-4 epithelial cells and RAW264.7 macrophages treated with PM_10_. PM_10_-conditioned medium from these LA-4/RAW264.7 cocultures was then applied to BV2 microglial cells to evaluate the impact of PM_10_-induced proinflammatory mediators on microglial activation. Finally, we investigated whether microglia stimulated with PM_10_-conditioned medium altered the level of neurotoxicity in neuroblastoma N2A cells.

## 2. Results

### 2.1. Chronic Intratracheal Instillation of PM_10_ Aggravated Motor Deficits in Mice with MPTP-induced PD

To investigate whether PM_10_ exposure contributes to the development and progression of Parkinson’s disease, mice received PM_10_ (5 mg/kg) twice weekly for 8 weeks before administration of the neurotoxin MPTP (Figure 1A). PM_10_-exposed mice without MPTP challenge displayed no significant change in motor function compared to PBS-treated mice, as demonstrated by the pole test evaluating bradykinesia [15] (Figure 1B,C) and the rotarod test measuring motor coordination [16] (Figure 1D). However, in the MPTP-treated mice, PM_10_ exposure resulted in pronounced motor deficits in the pole test (Figure 1B,C) and rotarod test (Figure 1D).

### 2.2. Chronic PM10 Exposure Enhanced the Degeneration of DA Neurons in PD Mice

Next, we analyzed whether PM_10_ exerts neurotoxic effects on DA neurons in PD. To test this hypothesis, we assessed tyrosine hydroxylase (TH) immunoreactivity to identify dopaminergic neurons in MPTP mice following chronic PM_10_ exposure (Figure 2A). As expected, compared to the PBS injection, MPTP administration led to a significant reduction in TH immunoreactivity in the pars compacta of the substantia nigra (SNpc) in the mice (Figure 2B,C). Importantly, the chronical PM_10_-exposed mice displayed a significant reduction in the TH-positive neurons of the SNpc in response to the MPTP challenge. There was no clear change in TH immunoreactivity in mice exposed to only PM_10_, suggesting that PM_10_ exposure (5 mg/kg) for 8 weeks alone does not induce notable DA neuron death. Next, we assessed the survival of DA neurons in the corpus striatum. Similar to the results from SNpc, PM_10_ exposure resulted in a decrease in TH immunoreactivity in the striatum of the MPTP-treated mice (Figure 2D). The neurotoxic effects of chronic PM_10_ exposure on MPTP-induced neurodegeneration were confirmed by western blotting analysis of the corpus striatum tissues (Figure 2E,F). 

### 2.3. Short-Term Exposure to PM_10_ Caused Microglial Activation

Despite numerous possibilities for how air pollution is linked to the brain, the pathologic effects of ambient particulate matter on neurodegeneration have been largely overlooked. Interestingly, a recent in vitro study demonstrated that diesel exhaust particles caused enhanced microglial activation, which subsequently induced selective damage to in vitro DA neurons [17]. Here, we hypothesized that PM_10_ exposure may be linked to microglia that accelerate the progression of Parkinson’s disease. To test this hypothesis, we exposed mice to PM_10_ (5 mg/kg) for a short period of time (5 days) and then evaluated microglial activation in the SNpc of mice by staining with ionized calcium-binding adaptor molecule 1 (IBA1), a marker of activated microglia (Figure 3A). PM_10_ exposure caused a significant increase in the number of cells positive for IBA1 compared to the control mice (Figure 3B,C). Furthermore, mice challenged with MPTP displayed enhanced IBA1^+^ cells in the substantia nigra compared to the PBS-treated control mice. Importantly, we observed that, following the MPTP challenge, acute PM_10_-exposed mice displayed enhanced IBA^+^ cells compared to mice exposed to only PM_10_ or treated to only MPTP.

### 2.4. Short-Term Exposure to PM_10_ Led to Pulmonary Inflammation

Considering the neurotoxic effect of PM_10_ exposure on neurodegeneration in PD, we hypothesized that intratracheal PM_10_ exposure may cause lung inflammation, which in turn can affect neuronal degeneration in the brain. To identify the direct impact of PM10 on mice, we exposed mice to PM_10_ (5 mg/kg) for a short period of time (5 days) (Figure 4A). Pulmonary inflammation was assessed by counting inflammatory cells from bronchoalveolar lavage (BAL) fluid. As expected, we observed a significant increase in total inflammatory cells in PM_10_-exposed mice compared to PBS-treated control mice (Figure 4B). High levels of inflammatory cells, including macrophages (Figure 4C), neutrophils (Figure 4D), and eosinophils (Figure 4E), were also detected in the lung tissues of the PM_10_-exposed group. Furthermore, following the MPTP challenge, the acute PM_10_-exposed mice displayed enhanced inflammatory cells in the lung tissues compared to the control mice. However, there was no clear difference in the number of inflammatory cells between PM_10_-treated mice and PM_10_-treated MPTP-challenged (PM_10_+MPTP) mice. Moreover, histological evaluations (Figure 4F) and semiquantitative analysis of inflammatory indices (Figure 4G) confirmed inflammatory cell infiltration and alveolar wall changes associated with lung inflammation, which were remarkably apparent in PM_10_ mice and PM_10_-exposed MPTP-treated mice.

### 2.5. Short-Term Exposure to PM_10_ Induced Systemic Inflammation in PD Mice

Considering PM_10_-induced inflammatory responses in the lungs, we asked whether intratracheal PM_10_ exposure causes systemic inflammation. To investigate PM_10_-induced systemic inflammation, proinflammatory cytokine levels in the serum of mice were assessed by enzyme-linked immunosorbent assay (ELISA) (Figure 5A). Importantly, we detected significant elevations in serum levels of interleukin-1β (IL-1β) (Figure 5B), tumor necrosis factor-α (TNF-α) (Figure 5C), and interleukin-6 (IL-6) (Figure 5D) in PM_10_-exposed mice compared to control mice. The levels of these investigated cytokines were not significantly different between the MPTP-challenged group and the control group, suggesting that systemic inflammation may not be significantly involved in the development of neurodegeneration in PD. Furthermore, there was no clear difference in the levels of IL-1β, TNF-α and IL-1β between PM_10_-treated mice and PM_10_-treated MPTP-challenged (PM_10_+MPTP) mice. 

### 2.6. PM_10_ Led to an Inflammatory Response in Cocultured Alveolar Epithelial Cells and Macrophages

Since the primary organs affected by PM_10_ instillation are the lungs, we hypothesized that PM_10_-induced neurotoxicity in mice is associated with inflammatory responses in the lungs. To test our hypothesis, we evaluated whether exposing cocultured lung epithelial cells and macrophages generates inflammatory responses in response to PM_10_. The inflammation was analyzed with quantitative real-time PCR of the cocultured LA-4/RAW264.7 cells (Figure 6A). The expression of proinflammatory cytokines and chemokines, such as IL-1β, chemokine (C-C motif) ligand 2 (CCL2), and TNF-α was significantly increased in cocultured LA-4/RAW264.7 cells in response to PM_10_ treatment (Figure 6B–D). Importantly, we observed that 50 μg/mL PM_10_ promptly induced IL-1β upregulation beginning at 2 h of treatment, which persisted for at least 24 h. (Figure 6B). Similarly, the levels of CCL2, and TNF-α mRNA were also increased significantly in cocultured LA-4/RAW264.7 cells in a time-dependent manner (Figure 6C,D). 

### 2.7. Conditioned Medium from PM_10_-Stimulated Cocultures of LA-4- and RAW264.7-Activated Microglial Cells

Next, we sought to investigate the potential inflammatory mechanisms by which PM_10_ induced neuroinflammation. We hypothesized that proinflammatory modulators released by the interaction of the lung epithelium and macrophages in response to PM_10_ can activate microglia. To test our hypothesis, we applied conditioned medium from cocultured LA-4/RAW264.7 cells following PM_10_ treatment (PM_10_-conditioned medium) to microglial BV2 cells (Figure 7A). Interestingly, 24 h of treatment with PM_10_-conditioned medium was able to activate microglial BV2 cells. BV2 cells treated with PM_10_-conditioned medium- exhibited an amoeba-like shape with short cellular processes (Figure 7B–D). The majority of the BV2 cells in PM_10_-conditioned medium were round to oval in shape, while control BV2 cells were ramified (multipolar or spindle-shaped). Additionally, we measured the mRNA levels of proinflammatory cytokines and chemokines to assess the inflammatory characteristics of BV2 cells treated with PM_10_-conditioned medium. BV2 cells treated with PM_10_-conditioned medium displayed significant upregulation of IL-1β and CCL2 mRNA compared with BV2 cells cultured in DMEM medium (Figure 7E). The mRNA level of TNF-α in BV2 cells with PM_10_-conditioned medium was also higher than that of DMEM medium. However, this finding did not reach statistical significance. 

### 2.8. Activated Microglia Induced by PM_10_-Conditioned Medium Aggravated Neuronal Toxicity against MPP^+^

Using an in vitro model, we further demonstrated the neurotoxic effect of PM_10_ in the inflammatory process. As expected, MPP^+^ treatment caused N2A cell death in a dose-dependent manner after 24 h of exposure (Figure 8A). As displayed in Figure 8B, we constructed a neuronal cell and microglial coculture model using murine neuroblastoma N2A cells and BV2 cells. In our system, BV2 cells were first activated with PM_10_-conditioned medium from PM_10_-treated LA-4 and RAW264.7 cells. After coculturing N2A and BV2, the neurotoxin 1-methyl-4-phenylpyridinium (MPP^+^) was added for 24 h. Interestingly, we observed cytotoxicity in N2A cells that were cocultured with activated BV2 cells by PM_10_-conditioned medium in the absence of MPP^+^ (Figure 8C). N2A cells clearly displayed a significant reduction in neuronal viability against MPP^+^ when the cells were cocultured with activated BV2 by PM_10_-conditioned medium. 

## 3. Discussion

PM_10_ is the most popular indicator of ambient air pollution [18]. While the association of PM_10_ exposure and human health problems is well described in the context of cardiovascular and respiratory diseases, it has only recently been shown that these harmful effects extend to the brain. To date, there is no clear evidence of the mechanism by which PM_10_ disrupts neuronal cells in Parkinson’s disease.

Recent epidemiologic studies have demonstrated an important association between PM exposure and neurodegeneration. Lilian et al. [5] determined that elderly people in cities with high levels of air pollution are susceptible to AD associated with neuroinflammation and amyloid-β accumulation in the cortex and hippocampus. Notably, long-term exposure to traffic-related air pollution aggravates AD-associated pathology [19]. Interestingly, recent studies have presented consistent evidence that chronic exposure to air pollutants elicits lung inflammation, with a clear increase in olfactory barrier disruption, systemic inflammation, and neuroinflammation [20]. Considering that PM_2.5_ easily and deeply penetrates the alveolar parenchyma, it is surprising that PM_10_ triggers stronger proinflammatory effects in the lung than PM_2.5_. These findings were reported in studies conducted in humans [21], human monocytes [22] and human alveolar macrophages [23]. These studies reported that proinflammatory cytokines and chemokines in the lungs, including TNF-α and IL-6, were significantly released by PM_10_ exposure. Moreover, PM_10_ exposure alone (1–3 months) was able to induce inflammatory gene expression in rat brains [24].

Consistent with prior studies, we observed pulmonary and systemic inflammation in mice following short-term PM_10_ exposure. Enhanced inflammatory cell infiltration was detected in the lungs of PM_10_-exposed control mice and PM_10_-exposed PD mice challenged with MPTP. In addition, upregulated serum inflammatory cytokines, such as TNF-α, IL-6, and IL-1β, were detected in PM_10_-exposed mice in both the PBS-treated and MPTP-treated groups. Additionally, the proinflammatory potential of PM_10_ was assessed in vitro in lung epithelial LA-4 and macrophage RAW264.7 cells. Indeed, the inflammatory response is a cascade of biological reactions that can be initiated by exposure to stimulants, including PM. In this response, the lung epithelium, together with macrophages, can contribute to the barrier function against inhaled materials, including PM. Considering the multitude of interactions taking place within the epithelium [25], inflammatory reactions mediated by epithelial cells may affect the function of many tissues in deleterious ways. Vignal et al. [26] found that 15 days of coarse PM exposure in mice resulted in severe inflammation of the colon and lung. The complex relationship between PM-induced lung inflammation and systemic or tissue-specific inflammation is still not sufficiently explained. However, many previous studies have focused on the production of a wide range of cytokines and chemokines, such as TNF-α, IL-8, CCL2, and IL-6, following PM exposure as a key mechanism affecting human health problems [25]. These proinflammatory cytokines and chemokines in the lungs produced in response to PM_10_ may then be released into the bloodstream, triggering local inflammatory responses associated with many diseases in diverse tissues. Indeed, recent epidemiological studies have provided insight into the many risks and inverse risk factors associated with PD. The association between PD and cancer discovered by epidemiological reports is particularly intriguing [27,28]. An increased risk of PD among patients with melanoma is well known [29,30]. The underlying etiology of this positive association between PD and melanoma remains disputable. Particulate matter may be one of the potential common risk factors between PD and melanoma because direct exposure to airborne pollutants, including PM, can induce skin cancer [31]. Considering that melanoma is an immunogenic tumor [32], immune dysfunction can be the possible underlying mechanism for these two distinct diseases. Further research is needed to elucidate the shared role of particulate matter in the pathogenesis of PD and cancer.

Although the neuropathology underlying PD is not well understood, several studies have consistently reported sustained and activated neuroinflammation in patients with PD and animal models of PD [33,34]. Indeed, while it has been overlooked for too long, recent studies have identified a link between PM_10_ and neurodegeneration in PD. Interestingly, the increasing prevalence of PD was found to be positively correlated with annual increases in airborne metals [35]. Moreover, Liu et al. [36] found a positive correlation between high levels of PM_10_ and PD risk in PD patients. These authors suggest that people predisposed to or suffering from PD may be especially concerned about brain health in highly polluted cities. Consistent with these studies, we found that long-term PM_10_-exposed PD mice displayed severe behavioral deficits and dopaminergic neuron death in the substantia nigra. Similar to the in vivo results, severe N2A neuronal cell death was observed in response to MPP^+^ treatment when the cells were cocultured with PM_10_-conditioned medium-induced activated microglia. Interestingly, we observed neuronal toxicity in N2A cells induced by microglia activated by PM_10_-conditioned medium even in the absence of MPP^+^ toxin. The neuronal toxicity induced by activated microglia has been well documented by LPS treatment [37,38]. Furthermore, Peters et al. [39] reported a significant reduction in TH-positive dopaminergic neurons in PM-exposed mice. To our knowledge, this is the first study to demonstrate the impact of PM_10_ on aggravating neuronal cell death in Parkinson’s disease by applying both in vivo and in vitro experiments. 

Our studies demonstrated that no difference between the control and PD mice was observed in the serum levels of IL-1β, TNF-α, and IL-6. These proinflammatory cytokines were not significantly altered in the PD mice following PM_10_ exposure compared to the PD group. However, the PD mice following PM_10_ exposure displayed higher levels of proinflammatory cytokines in the serum after PM_10_ exposure. In Parkinson’s disease, neuroinflammation is mainly associated with microglial activation, which can underlie neurodegenerative pathology. Beyond neuroinflammation, recent evidence suggests systemic inflammation, with an ongoing immune response in the brain, is a potential driving factor driving neurodegeneration in PD [40,41]. Systemic inflammation induced by chronic IL-1β exacerbated neurodegeneration and microglial activation in the substantia nigra of a 6-hydroxydopamine (6-OHDA)-induced mouse model of PD [42]. Indeed, it is surprising that long-term use of non-steroidal anti-inflammatory drugs (NSAIDs) produced protective effects against Parkinson’s disease in human [43]. It is likely that inflammatory mediators that are produced in the periphery during neuronal loss in PD may spark off exacerbation in the neurodegeneration. In recent years, it has become accepted that α-synuclein has a key role in microglia-mediated neuroinflammation, which accompanies the development of α-synucleinopathy, such as Parkinson’s disease and dementia with Lewy Bodies. More recently, some studies have reported a link between microglia and tau in atypical Parkinsonian syndromes, such as progressive supranuclear palsy (PSP) [44]. Despite the impact of neuroinflammation on neurodegeneration, the role of microglial activation in different Parkinsonian syndromes remains unclear. Specific blood parameters, such as neutrophil-to-lymphocyte ratio (NLR) and platelet-to-lymphocyte ratio (PLR), have been proposed as diagnostic and predictive markers reflecting distinct inflammatory features to distinguish α-synucleinopathies and taupathies clinically. A better understanding of the link between systemic inflammation and PD neurodegeneration and related pathomechanism is essential.

How does exposure to PM_10_ exacerbate neuronal death in Parkinson’s disease? The recently presented lung–brain axis hypothesis suggested the role of microglia in affecting brain health problems through air pollution-induced inflammatory circulating factors [45]. PM_10_ exposure-induced proinflammatory mediators may likely be released into the blood and then disrupt the blood–brain barrier (BBB) and subsequently localize in the brain regions involved in PD. Under this systemic inflammation induced by PM_10_, additional cytokines and inflammatory cells can further enter the brain via the impaired BBB to enhance neuroinflammation. This may explain why residents in polluted areas displayed severe neuroinflammation and a brain innate immune response corresponding to BBB damage [5,6,46]. The possible mechanism of PM_10_-induced neurotoxicity in PD is presented in the Graphical Abstract.

In summary, our results demonstrate that PM_10_ exerts a significant deleterious effect on the neurodegeneration of Parkinson’s disease. Our data show that PM-induced toxicity is associated with enhanced pulmonary and systemic inflammatory responses. In response to PM_10_, systemic inflammation appears to exacerbate microglia-mediated neuroinflammation and subsequently accelerate the neurodegenerative features of Parkinson’s disease. Together with known environmental risk factors, our findings can facilitate therapeutic interventions targeting PM_10_ exposure and neurotoxic susceptibility against Parkinson’s disease.

## 4. Materials and Methods

### 4.1. Animals

Seven- to eight-week-old male C57BL/6J mice were purchased from the Jackson Laboratory (Bar Harbor, ME, USA). All male mice used in the experiment were maintained under pathogen-free conditions on a 12 h light/dark cycle and temperature-controlled conditions, with food and water provided ad libitum. All experiments were carried out in accordance with approved animal protocols and guidelines established by Kyung Hee University and Dong-A University.

### 4.2. MPTP Intoxication

A mouse model of PD was generated using the neurotoxin MPTP, as previously described [47]. Briefly, MPTP (20 mg/kg; Sigma, St. Louis, MO, USA) was administered intraperitoneally (i.p.) to mice four times a day at 2-h intervals, inducing severe and persistent depletion of dopamine. During the experiment, the mice were monitored for their physical condition and weight loss. The mortality rate of mice following MPTP injection was less than 30% in each group. The mice surviving with <20% weight loss were included in the study.

### 4.3. Intratracheal Instillation of PM_10_

Here, we used commercially available course urban dust particles that resemble PM_10_ collected from urban settings. These particles do not contain any transition metals (e.g., Cu) or aromatic organic compounds on particles (e.g., polycyclic hydrocarbons (PAHs)), which can induce serious health hazards and increase the risk of cancer [48]. These dust particles, which were obtained from European Reference Materials (ERM; reference code ISO 170345), are widely used to simulate airborne PM_10_ [49,50,51]. Table 1 displays the constituents of the PM_10_ used in this study. We intratracheally administered PM_10_ or phosphate buffered saline (PBS) to mice for 5 days (acute exposure) or 8 weeks (chronic exposure), as previously described, with minor modifications [52]. Briefly, the mice were positioned with their tongue gently extended. Then, 5 mg/kg PM_10_ in 50 µL of PBS was placed at the back of the oral cavity and injected into the trachea. Immediately prior to instillation, PM_10_ was vortexed for 30 s to suspend the particles. After instillation, the mice were allowed to recover under visual inspection before returning to their cages.

### 4.4. Immunohistochemistry

After the mice were sacrificed, transcardial perfusion was performed with PBS and 4% paraformaldehyde in PBS, and the brains were immediately removed. The samples were fixed overnight at 4 °C with 4% paraformaldehyde in PBS and stored in a 30% sucrose solution at 4 °C until they sank. Coronal sections (thickness, 30 µm) were cut using a sliding microtome. The sections were incubated with primary antibodies overnight at 4 °C. The following primary antibody was used: anti-tyrosine hydroxylase (TH; 1:1000, Pel-Freez Clinical Systems, Brown Deer, WI, USA). After washing with PBS, a secondary biotinylated antibody in an avidin-biotin complex kit (Vectastain ABC kit; Vector Laboratories, Burlingame, CA, USA) was added to the sections for 1 h, following the manufacturer’s instructions. Images of the stained sections were obtained using a light microscope (Nikon, Tokyo, Japan). 

### 4.5. Unbiased Stereological Estimation

Unbiased stereological estimation was carried out using an optical fractionator on an Olympus CAST (computer-assisted stereological toolbox system) system, version 2.1.4 (Olympus, Ballerup, Denmark), as previously described [53,54]. The sections used for counting TH- or IBA1-positive cells covered the entire SN from the rostral tip of the pars compacta to the caudal end of the pars reticulate (anteroposterior, −2.06 to −4.16 mm from the bregma). The counting frame was placed randomly on the first counting area and moved systematically over all counting areas until the entire delineated area was sampled. The total number of stained cells was estimated according to the optical fractionator equation [55].

### 4.6. Analysis of Bronchoalveolar Lavage Fluid

Bronchoalveolar lavage (BAL) fluid was collected, as previously described [56]. Briefly, BAL fluid was collected by washing with 1 mL of PBS three times. The samples were then centrifuged, and the pellets were resuspended. The total number of cells was quantified using a hemocytometer under bright-field microscopy. Furthermore, the cells were stained with a Diff-Quick staining kit (Thermo Fisher Scientific, Waltham, MA, USA).

### 4.7. Locomotor Activity Measurement

To assess bradykinesia, the pole test was performed as previously described [57]. The time to orient downward and the time to descend the pole were recorded. This test was used to measure motor performance five times, and the average duration of the 3 best performances was recorded. Mice were allowed to rest for 5 min to avoid stress and fatigue. Motor coordination was measured using the rotarod test, as previously described [58]. The rotation speed was increased from 4 to 40 rpm over 5 min. The latency to fall was measured for each animal. Each mouse was tested three times, and the median time was calculated.

### 4.8. Western Blotting Analysis

Loss of TH protein in the striatum of MPTP-treated mice was assessed. After extracting the brain tissues, coronal sections of the striatum within a brain matrix (RWD Life Science Inc., Shenzhen, China) were homogenized with a glass homogenizer in 50 mM Tris-HCl, pH 7.4, 100 mM NaCl, and 1% SDS with protease and phosphatase inhibitor cocktails (Sigma, St. Louis, MO, USA). After protein preparation, protein concentrations were determined with a BCA Protein Assay Kit™ (ThermoFisher Scientific, Waltham, MA, USA). The samples were subjected to electrophoresis, transferred, and immunoblotted with antibodies against TH (1:1000, Pel-Freez Clinical Systems, Brown Deer, WI, USA) or beta-actin (1:500, Santa Cruz Biotechnology, Santa Cruz, CA, USA). The signals were visualized with enhanced chemiluminescence reagents (Clarity Western ECL Substrate, Bio–Rad, Hercules, CA, USA). The results were analyzed using the ImageJ densitometry system.

### 4.9. Enzyme-Linked Immunosorbent Assay

Levels of the inflammatory cytokines TNF-α, IL-1β and IL-6 in the serum of the blood of the mice were quantified using a quantitative sandwich ELISA according to the manufacturer’s instructions (BD Biosciences, San Diego, CA, USA). The optical density (OD) of each sample after color development at 450 nm was quantified with a microplate reader (SoftMax PRO software, Sunnyvale, CA, USA).

### 4.10. Quantitative Real-Time Polymerase Chain Reaction (Real-Time qPCR)

Total RNA was extracted from lung tissues and cells using TRIzol (Thermo Fisher Scientific, Waltham, MA, USA). cDNA was synthesized with a cDNA synthesis kit (Bioneer Corporation, Daejeon, Korea) according to the manufacturer’s instructions. Real-time PCR analysis was performed in a LightCycler 96 (Roche, Basel, Switzerland) with SYBR Green I master mix using a SensiFAST™ SYBR^®^ No-ROX Kit (Bioline, Paris, France). The oligonucleotides used are shown in Table 2.

### 4.11. Cell Culture and PM_10_ Stimulus

Murine RAW 264.7 macrophages, LA-4 lung epithelial cells, and BV-2 microglia were purchased from the American Type Culture Collection (ATCC, Manassas, VA, USA). In a RAW264.7/LA-4 coculture system, LA-4 cells were seeded until they completely adhered. After 2 days, RAW264.7 cells were plated at a ratio of 1:10 (RAW264.7:LA-4). The next day, 50 μg/mL of PM_10_ was applied to the cocultured LA-4/RAW264.7 cells for 24 h. These media were applied to BV2 to induce microglial activation.

### 4.12. Cytotoxicity Assay

In a coculture of microglia and neurons, N2A cells were grown in the bottom of wells. BV2 cells were grown in culture inserts (pore size 0.4 μm; Corning, New York, NY, USA) and activated with PM_10_-conditioned medium as described above. Then, these BV2 in culture inserts were placed on top of the N2A-containing wells. MPP^+^ (Sigma, St. Louis, MO, USA) was applied to the culture bottom for 24 h. To assess neuronal toxicity against MPP^+^, MTT (5 mg/mL; Sigma, St. Louis, MO, USA) was added. After incubating for 4 h, the formazan crystals were solubilized with SDS solution (10% SDS and 0.01 M HCl). Absorbance was detected using a multi-plate reader at a wavelength of 570 nm (A_570_). 

### 4.13. Statistics

Statistical analysis was performed using GraphPad Prism (v5.0; GraphPad, San Diego, CA, USA) software. Normal distribution and homogeneity of variance were analyzed. Statistical significance was then determined by one-way analysis of variance (ANOVA), followed by Tukey’s post-hoc multiple comparison test or nonparametric Kruskal–Wallis analysis when applicable. Data are expressed as the mean ± standard error of measurement (SEM). *p* < 0.05 was considered to indicate statistical significance.

## Figures and Tables

**Figure 1 ijms-23-06487-f001:**
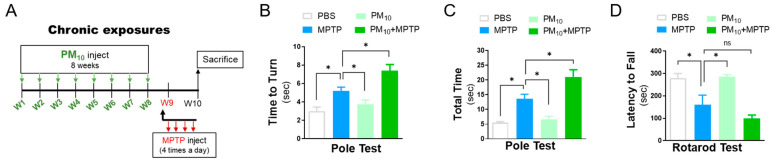
Long-term exposure to PM_10_ enhanced the severity of behavioral deficits in MPTP-treated mice. (**A**) Experimental schematic describing long-term exposure to PM_10_ (5 mg/kg) for 8 weeks and the MPTP-induced mouse model of Parkinson’s disease used to assess motor impairments in mice. (**B**–**D**) Motor impairment was assessed by the pole test (**B**,**C**) and rotarod test (**D**) 1 week post-MPTP challenge. The data are expressed as the mean ± SEM (*n* = 5 per group). * *p* < 0.05. ns indicates no significant differences among the groups.

**Figure 2 ijms-23-06487-f002:**
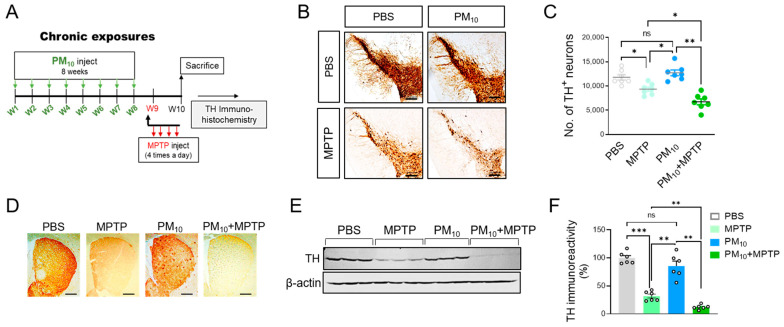
Long-term exposure to PM_10_ aggravated the loss of dopaminergic neurons in MPTP mice. (**A**) Experimental schematic describing long-term exposure to PM_10_ (5 mg/kg) for 8 weeks and the MPTP-induced mouse model of PD to measure dopaminergic neuron death. (**B**) Representative images of TH-stained dopaminergic neurons in the pars compacta of the substantia nigra (SNpc). (**C**) Stereological counts of TH-positive neurons in SNpc. (**D**) Representative images of TH-stained dopaminergic neurons in the striatum. (**E**) Western blotting analysis of TH in the striatum. (**F**) Quantitative analysis of TH protein in the striatum, measured by western blotting. The protein level of TH was calculated compared to β-actin was calculated. Quantified data were normalized to the control group (the mean value of the control group was equal to 1). The scale bar is 100 μm. Data are shown as the mean ± SEM (*n* = 6 per group). * *p* < 0.05; ** *p*  <  0.01; *** *p* < 0.001. ns = no significant difference among the groups.

**Figure 3 ijms-23-06487-f003:**
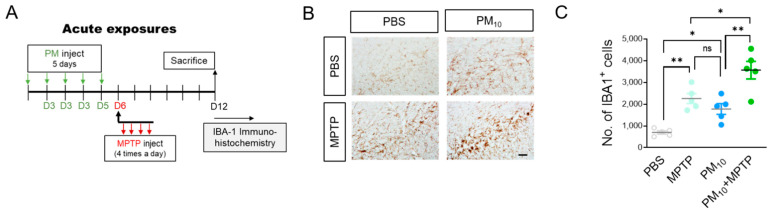
Short-term exposure to PM_10_ induced microglial activation in mice. (**A**) Experimental schematic describing short-term exposure to PM_10_ (5 mg/kg) for 5 days and PD mice following MPTP challenge to evaluate microglial activation in the pars compacta of the substantia nigra. (**B**,**C**) Representative images of IBA1-positive cells (**B**) are shown, and the level of immunoreactivity (**C**) was calculated with the presented mean values. The scale bar is 30 μm. Data are shown as the mean ± SEM (*n* = 5 per group). * *p* < 0.05; ** *p*  <  0.01. ns = no significant difference among the groups.

**Figure 4 ijms-23-06487-f004:**
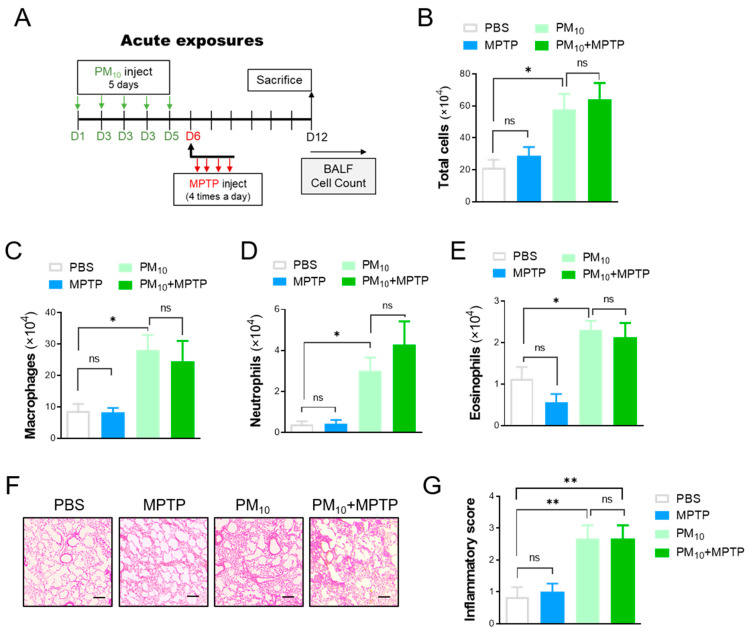
Short-term exposure to PM_10_ induced pulmonary inflammation in mice. (**A**) Experimental schematic describing short-term exposure to PM_10_ (5 mg/kg) for 5 days and PD mice following MPTP challenge to evaluate lung inflammation. (**B**–**E**) Bronchoalveolar large (BAL) fluid was analyzed by counting the total number of inflammatory cells (**B**), macrophages (**C**), neutrophils (**D**), and eosinophils (**E**) in mice. (**F**,**G**) Representative images of lung tissues stained with hematoxylin and eosin (H&E). The scale bar is 50 μm. The data are expressed as the mean ± SEM (*n* = 5–8 per group). * *p* < 0.05; ** *p*  <  0.01. ns indicates no significant differences among the groups.

**Figure 5 ijms-23-06487-f005:**
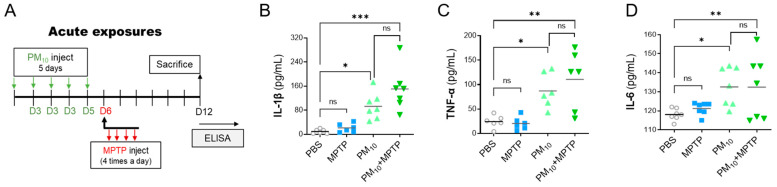
Short-term exposure to PM_10_ triggered systemic inflammation in mice. (**A**) Experimental schematic describing short-term exposure to PM_10_ (5 mg/kg) for 5 days and PD mice following MPTP challenge to investigate systemic inflammation. (**B**–**D**) The levels of proinflammatory cytokines in the serum of mice were analyzed by ELISA. The investigated cytokines were IL-1β, TNF-α, nd IL-6. The data are expressed as the mean ± SEM (*n* = 5–7 per group). * *p* < 0.05; ** *p*  <  0.01; *** *p* < 0.001. ns indicates no significant differences among the groups.

**Figure 6 ijms-23-06487-f006:**
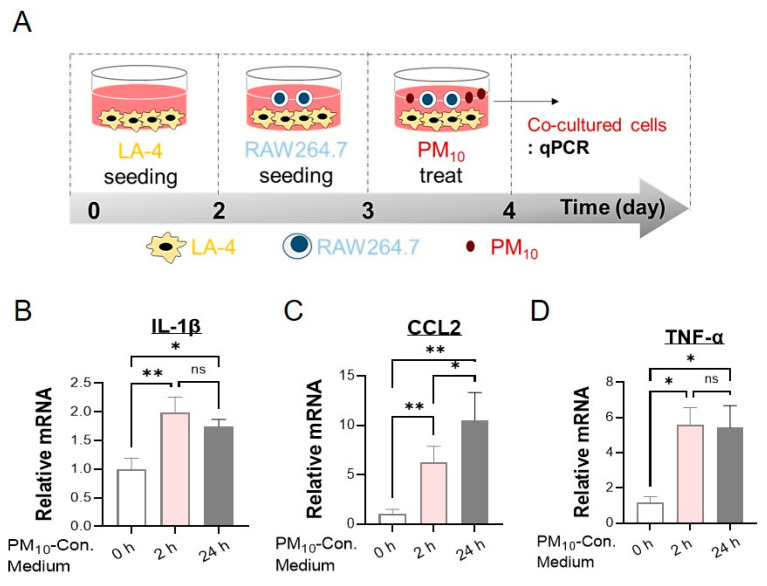
Inflammatory effect of PM_10_ in cocultured LA-4/RAW264.7 cells. (**A**) In vitro schematic displaying the coculture system with lung epithelial LA-4 cells and macrophages against PM_10_ stimulation. PM_10_ (50 μg/mL) was applied to cocultured cells for 0, 2, and 24 h. (**B**–**D**) Real-time qPCR analysis was used to measure the mRNA levels of proinflammatory cytokines and chemokines, including IL-1β (**B**), CCL2 (**C**), and TNF-α (**D**), in LA-4/RAW264.7 cells. Data are presented as the mean ± SEM of five independent experimental repeats. The relative level of proinflammatory cytokines was normalized to that of β2-microglobulin (B2M) and calculated as the fold change compared to those in the PBS-treated group. * *p* < 0.05; ** *p*  <  0.01. ns = no significant difference among the groups.

**Figure 7 ijms-23-06487-f007:**
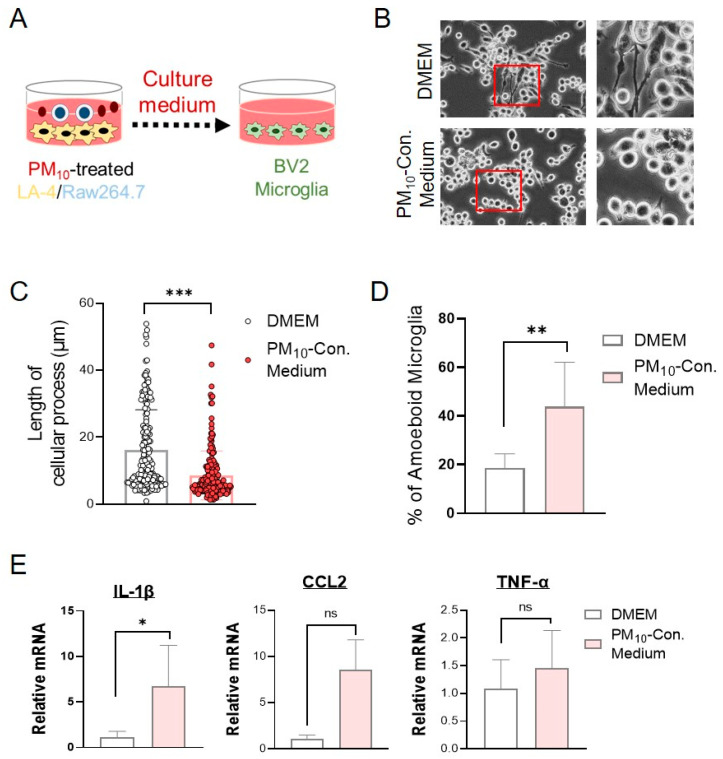
Microglial activation induced by PM_10_-conditioned medium from cocultured LA-4/RAW264.7 cells. (**A**) Experimental schematic describing the application of PM_10_-conditioned medium from LA-4/RAW264.7 cells cocultured into BV2 microglial cells. The culture medium was obtained from cocultured LA-4/RAW264.7 after 24 h of PM_10_ treatment (50 μg/mL). The PM_10_-conditioned medium was then applied to BV2 cells for 24 h. (**B**) Morphological changes in BV2 cells following the application of PM_10_-conditioned medium were observed under a phase-contrast microscope (shown at 400× magnification). (**C**) Cellular processes of microglial BV2 cells were measured with ImageJ using a NeuronJ plugin. The path length from the soma to the point where the process terminates was calculated and counted using the neurite tracing method. (**D**) Activated microglia showing an amoeboid shape were counted under a phase-contrast microscope. The results are expressed as the percentage of activated cells. (**E**) Real-time qPCR analysis was performed to measure the mRNA levels of proinflammatory cytokines and chemokines, such as IL-1β, CCL2, and TNF-α, in PM_10_-conditioned medium-stimulated BV2 microglia. The data are expressed as the mean ± SEM (*n* = 4–5 per group). * *p* < 0.05; ** *p*  <  0.01; *** *p* < 0.001. ns = no significant difference among the groups.

**Figure 8 ijms-23-06487-f008:**
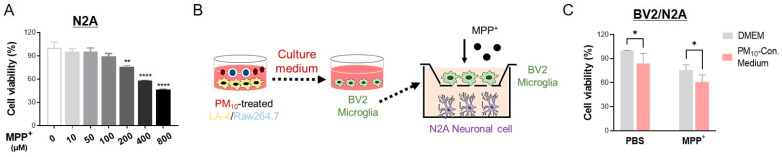
The neurotoxicity of MPP^+^ to N2A cells was enhanced by microglia activated with PM_10_-conditioned medium. (**A**) Cell viability was analyzed using an MTT assay to investigate the dose-dependent toxicity of MPP^+^ on N2A cells cultured alone. (**B**) Experimental schematic describing the coculture system of neuronal N2A cells and BV2 microglia activated by PM_10_-conditioned medium using Transwell chambers (pore size 0.4 μm). The medium was first collected from cocultured of LA-4/RAW264.7 cells following 24 h of PM_10_ exposure. The PM_10_-conditioned medium was then applied to BV2 cells for 24 h to activate microglia. N2A cells were plated on the bottom, and then activated BV2 cells were plated on the top of the Transwell inserts. These N2A and BV2 cells were cultured in DMEM (without any PM_10_ stimulus) with Neurotoxin MPP^+^ for 24 h. (**C**) N2A cells cocultured with microglia activated by PM_10_-conditioned medium were treated with MPP^+^ (200 μM) for 24 h. N2A cell survival was determined by MTT analysis. The value was normalized to each non-MPP^+^ treated group. The data are expressed as the mean ± SEM (*n* = 4–5 per group). * *p* < 0.05; ** *p* < 0.01; **** *p* < 0.0001.

**Table 1 ijms-23-06487-t001:** Elemental constituents of the PM_10_ used in this study.

Element	Mass Fraction	
	Certified Value (mg/kg) ^(1)^	Certified Uncertainty ^(2)^ (mg/kg)
Arsenic	7.1	0.7
Cadmium	0.90	0.22
Lead	113	17
Nickel	58	7

^(1,2)^ The certified values and their uncertainties are mass fractions based on the mass of the sample. ^(2)^ The certified uncertainty is the expanded uncertainty with a coverage factor *k* = 2, corresponding to a level of confidence of 95%, estimated in accordance with the ISO/IEC Guide.

**Table 2 ijms-23-06487-t002:** Oligonucleotides used in real-time qPCR.

Gene	Direction	Sequence (5′-> 3′)
TNF-α	Forward	gcctcttctcattcctgcttg
	Reverse	ctgatgagagggaggccatt
IL-1β	Forward	cacagcagcacatcaacaag
	Reverse	gtgctcatgtcctcatcctg
CCL2	Forward	tgatcccaatgagtaggctggag
	Reverse	atgtctggacccattccttcttg
B2M	Forward	ctgctacgtaacacagttccaccc
	Reverse	catgatgcttgatcacatgtctcg

## Data Availability

The raw data of the analyses presented in the figures in this study are available upon request from the corresponding author.

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
