# Peer review of "Particulate Matter Exacerbates the Death of Dopaminergic Neurons in Parkinson’s Disease through an Inflammatory Response"

_ijms, 2022, doi:10.3390/ijms23126487_

Round 1

Reviewer 1 Report

The significance of particulate matter in the pathogenesis of neurodegenerative disorders seems crucial, especially when evaluating it in correlation with the analysis of neuroinflammation in this group of diseases. Authors of the study by Bae et al elaborate on this issue in the context of particulate matter, a challenge of our times. I have several points which should be addressed before further consideration:

  1. Particulate matter and Parkinson's disease and particulate matter were found to be risk factors of cancer. Interestingly PD is a risk factor for some cancers [A,B]. I think that it would be valuable if authors could speculate whether in their opinion the combined risk factors could additionally exacerbate this form of pathogenesis. ( A - Association Between Parkinson's Disease and Melanoma: Putting the Pieces Together. Front Aging Neurosci. 2020 Mar 10;12:60. doi: 10.3389/fnagi.2020.00060. PMID: 32210791; PMCID: PMC7076116., B - The Links between Parkinson's Disease and Cancer. Biomedicines. 2020 Oct 14;8(10):416. doi: 10.3390/biomedicines8100416. PMID: 33066407; PMCID: PMC7602272.)
  2. Authors associate the "neurodegenerative" role with microglial activation, however I think it would be important to stress that this process was found to be an enhacing factor not only in synucleinopathies but also tauopathic atypical parkinsonian syndromes. To provide a clear overview to the readership of the journal I think that authors should stress the potential role of microglial activation in both group of this diseases and give their point of view why in certain cases the evolution leads to synucleinopathies, whereas in other to tauopathies as progressive supranuclear palsy or corticobasal syndrome. Authors should refer to recent publications referring to neuroinflammatory perspective in both diseases e.g. C - The neutrophil-to-lymphocyte ratio as a marker of peripheral inflammation in progressive supranuclear palsy: a retrospective study. Neurol Sci. 2020 May;41(5):1233-1237. doi: 10.1007/s10072-019-04208-4. Epub 2020 Jan 4. PMID: 31901125., D - Platelet-to-lymphocyte ratio and neutrophil-tolymphocyte ratio may reflect differences in PD and MSA-P neuroinflammation patterns. Neurol Neurochir Pol. 2022;56(2):148-155. doi: 10.5603/PJNNS.a2022.0014. Epub 2022 Feb 4. PMID: 35118638., E - Neutrophil-to-lymphocyte ratio (NLR) at boundaries of Progressive Supranuclear Palsy Syndrome (PSPS) and Corticobasal Syndrome (CBS). Neurol Neurochir Pol. 2021;55(1):97-101. doi: 10.5603/PJNNS.a2020.0097. Epub 2020 Dec 14. PMID: 33315235.
  3. An additional paragraph elaborating on future perspectives concerning this finding would be beneficial.

Author Response

Response to Reviewer #1

We would like to thank the reviewer for careful and thorough reading of this manuscript and also for thoughtful comments and constructive suggestions, which help to improve the quality of this manuscript.

We agree with the reviewer’s suggestion and have rewritten discussion part so that it is now clearer and not make misleading or confusing. We acknowledge for the reviewer’s comments and have added future perspective in the last part of discussion. We hope that the reviewer finds these changes and this comply with the reviewer’s remarks.

Comment 1.      Particulate matter and Parkinson's disease and particulate matter were found to be risk factors of cancer. Interestingly PD is a risk factor for some cancers [A,B]. I think that it would be valuable if authors could speculate whether in their opinion the combined risk factors could additionally exacerbate this form of pathogenesis. ( A - Association Between Parkinson's Disease and Melanoma: Putting the Pieces Together. Front Aging Neurosci. 2020 Mar 10;12:60. doi: 10.3389/fnagi.2020.00060. PMID: 32210791; PMCID: PMC7076116., B - The Links between Parkinson's Disease and Cancer. Biomedicines. 2020 Oct 14;8(10):416. doi: 10.3390/biomedicines8100416. PMID: 33066407; PMCID: PMC7602272.)

Response 1.        We thank the reviewer for bringing up this point. We have accordingly rewritten sections in the discussion (Page 9, Line 314-323).

“Indeed, recent epidemiological studies have provided insight into many risks and inverse risk factors associated with PD. The association between PD and cancer discovered by epidemiological reports is particularly intriguing [27, 28]. An increased risk of PD among patients with melanoma is well known [29, 30]. The underlying etiology of this positive association between PD and melanoma is still disputable. Particulate matter may be one of potential common risk factors between PD and melanoma because direct exposure to airborne pollutants including PM can induce skin cancer [31]. Considering melanoma is an immunogenic tumor [32], immune dysfunction can be the possible underlying mecha-nisms for these two distinct diseases. Further research needs to elucidate the shared role of particulate matter in the pathogenesis of PD and cancer”.

Comment 2.      Authors associate the "neurodegenerative" role with microglial activation, however I think it would be important to stress that this process was found to be an enhacing factor not only in synucleinopathies but also tauopathic atypical parkinsonian syndromes. To provide a clear overview to the readership of the journal I think that authors should stress the potential role of microglial activation in both group of this diseases and give their point of view why in certain cases the evolution leads to synucleinopathies, whereas in other to tauopathies as progressive supranuclear palsy or corticobasal syndrome. Authors should refer to recent publications referring to neuroinflammatory perspective in both diseases e.g. C - The neutrophil-to-lymphocyte ratio as a marker of peripheral inflammation in progressive supranuclear palsy: a retrospective study. Neurol Sci. 2020 May;41(5):1233-1237. doi: 10.1007/s10072-019-04208-4. Epub 2020 Jan 4. PMID: 31901125., D - Platelet-to-lymphocyte ratio and neutrophil-tolymphocyte ratio may reflect differences in PD and MSA-P neuroinflammation patterns. Neurol Neurochir Pol. 2022;56(2):148-155. doi: 10.5603/PJNNS.a2022.0014. Epub 2022 Feb 4. PMID: 35118638., E - Neutrophil-to-lymphocyte ratio (NLR) at boundaries of Progressive Supranuclear Palsy Syndrome (PSPS) and Corticobasal Syndrome (CBS). Neurol Neurochir Pol. 2021;55(1):97-101. doi: 10.5603/PJNNS.a2020.0097. Epub 2020 Dec 14. PMID: 3331523

Response 2.        We totally agree with the reviewer. In response to this comment, we have rewritten discussion section accordingly. (Page 10, Line 357-368).

“In recent years, it has become accepted that α-synuclein has a key role in the microglia-mediated neuroinflammation, which accompanies the development of α-synucleinopathy, such as Parkinson’s disease and dementia with Lewy Bodies. More recently, there are some studies reporting the link between microglia and tau in atypical parkinsonian syndromes, such as progressive supranuclear palsy (PSP) [44]. Despite the impact of neuroinflammation on neurodegeneration, the role of microglial activation in different parkinsonian syndromes still remains unclear. Specific blood parameters, such as neutrophil-to-lymphocyte ratio (NLR) and platelet-to-lymphocyte ratio (PLR), have been proposed as diagnostic and predictive markers reflecting distinct inflammatory features to distinguish α-synucleinopathies and taupathies clinically. A better understanding of the link between systemic inflammation and PD neurodegeneration and related pathomechanism is essential.”

Comment 3.      An additional paragraph elaborating on future perspectives concerning this finding would be beneficial

Response 3.        We agree about this point. We thank you for this suggestion. We have accordingly rewritten sections in the discussion (Page 11, Line 380-387).

                           “In summary, our results demonstrate that PM10 exerts a significant deleterious effect on neurodegeneration of Parkinson’s disease. Our data show that PM induced toxicity is associated with enhanced pulmonary and systemic inflammatory response. In response to PM10, systemic inflammation appears to exacerbate microglia-mediated neuroinflammation and subsequently accelerate neurodegenerative features of Parkinson’s disease. Together with the known environmental risk factors, our findings can facilitate therapeutic intervention targeting PM10 exposure and neurotoxic susceptibility against Parkinson’s disease.”

Reviewer 2 Report

This manuscript by Choi et al focuses on understanding the role of PM10 on the progression of neurodegeneration during PD. Overall, this manuscript is well presented and the experiments are logical. However, there are a few concerns that will need to be addressed before this manuscript is suitable for publication. Points to be addressed are listed below in point-wise fashion:

(1) For Figures 1, 2 and 3, please present that data as individual data points per condition, so that the spread of individual mouse data points can be visualized. This is important because the data could be skewed or there could be a clustering of data points that is not visible in the way the data are currently presented.

(2) Please specify the sex of mice. Are the mice a mixture of males and females?

(3) Major point: The in vitro experiments are informative, but do not really help in understanding if PM10 exposure causes increased microglial activation in vivo. The authors need to immunostain midbrain and striatal sections of control, MPTP and MPTP + PM10 treated mice for a microglial marker such as IBA1 and assess microglial reactivity in the midbrain and in the striatum using a quantitative method as is done in vitro in Figure 6C and 6D. Without these data, the conclusion that PM10 mediates effects via microglial activation is not convincing. 

(4) Data in Figure 4B are not statistically significant between MPTP and MPTP + PM10 mice. However, the conclusion presented in lines 159 - 160 that systemic inflammation does not play a role in the effect of PM10 is incorrect and not supported by the data. Raised inflammation due to PM10 could still play a role in accelerating neurodegeneration and this does not have to be higher in intensity than MPTP only treated mice for the inflammation to have a detrimental effect. Please reword the writing in lines 159 - 160, and please also revise the discussion to include this concept, along with references pointing to a role for systemic inflammation in PD.

(5) The methods section does not have a description of the stereology method used to quantify TH+ neurons in the midbrain. Please write out a section on stereology methodology in methods.

Author Response

Response to Reviewer #2

We would like to thank the reviewer for careful and thorough reading of this manuscript. We acknowledge for the reviewer’s comments and have added new data about microglial activation in vivo experiment as figure 3. So in the revised version, total number of figures is 8 except graphic abstract. We also have rewritten discussion part in response to reviewer’s suggestion.

We hope that the reviewer finds these changes and this comply with the reviewer’s remarks.

Comment 1.      For Figures 1, 2 and 3, please present that data as individual data points per condition, so that the spread of individual mouse data points can be visualized. This is important because the data could be skewed or there could be a clustering of data points that is not visible in the way the data are currently presented.

Response 1.        We agree about this point. We thank you for this suggestion. We changed figure graph type in figure 1, 2, 3, and 4 accordingly.

Comment 2.      Please specify the sex of mice. Are the mice a mixture of males and females?

Response 2.        Thank you so much for pointing out. We have rewritten method section accordingly. (Page 11, Line 391-394)

“Seven- to eight-week-old male C57BL/6J mice were purchased from The Jackson Labora-tory (Bar Harbor, ME, USA). All male mice used in the experiment were maintained under pathogen-free conditions on a 12 h light/dark cycle and temperature-controlled conditions, with food and water provided ad libitum”

Comment 3.      (3) Major point: The in vitro experiments are informative, but do not really help in understanding if PM10 exposure causes increased microglial activation in vivo. The authors need to immunostain midbrain and striatal sections of control, MPTP and MPTP + PM10 treated mice for a microglial marker such as IBA1 and assess microglial reactivity in the midbrain and in the striatum using a quantitative method as is done in vitro in Figure 6C and 6D. Without these data, the conclusion that PM10 mediates effects via microglial activation is not convincing.

Response 3.        We thank you for this suggestion. We have now added new figure with staining IBA1 and counting IBA1+ cells on mice responding to reviewer’s comment and have put this data in Figure 3 as follows.

Comment 4.      Data in Figure 4B are not statistically significant between MPTP and MPTP + PM10 mice. However, the conclusion presented in lines 159 - 160 that systemic inflammation does not play a role in the effect of PM10 is incorrect and not supported by the data. Raised inflammation due to PM10 could still play a role in accelerating neurodegeneration and this does not have to be higher in intensity than MPTP only treated mice for the inflammation to have a detrimental effect. Please reword the writing in lines 159 - 160, and please also revise the discussion to include this concept, along with references pointing to a role for systemic inflammation in PD.

Response 4.        We thank you for this comment and apologized for this. We have rewritten this sentence in result section as follows. (Page 5-6, Line 180-182)

“Furthermore, there was no clear difference in the levels of IL-1β, TNF-α and IL-1β between PM10-treated mice and PM10-treated MPTP-challenged (PM10+MPTP) mice”

                           In response to reviewer’s comment, we have also rewritten discussion section accordingly as follows. (Page 10, Line 343-356).

“Our studies demonstrated that no difference between control and PD mice were observed in the serum levels of IL-1β, TNF-α, and IL-6. These proinflammatory cytokines were not significantly altered in PD mice following PM10 exposure compared to PD group. However, PD mice following PM10 exposure displayed higher levels of proinflammatory cytokines in the serum after PM10 exposure. In Parkinson’s disease, neuroinflammation has known to be mainly associated with microglial activation that can underlie the neurodegenerative pathology. Beyond neuroinflammation, recent evidence suggested systemic inflammation, with ongoing immune response in the brain, as a potential driving factor driving the neurodegeneration in PD [40, 41]. Systemic inflammation induced by chronic IL-1β exacerbated neurodegeneration and microglial activation in the substantia nigra of 6-hydroxydopamine (6-OHDA)-induced mouse model of PD [42]. Indeed, it is surprising that long-term use of non-steroidal anti-inflammatory drugs (NSAIDs) produced protective effects against Parkinson’s disease in human [43]. Probably, inflammatory mediators that produced in the periphery”

Comment 5.      The methods section does not have a description of the stereology method used to quantify TH+ neurons in the midbrain. Please write out a section on stereology methodology in methods

Response 5.        We thank you for this comment. We have rewritten method section and added this part accordingly (Page 12, Line 340-349).

“4.5. Unbiased stereological estimation

Unbiased stereological estimation was carried out using an optical fractionator on an Olympus CAST (computer-assisted stereological toolbox system) system, version 2.1.4 (Olympus, Ballerup, Denmark), as we previously described [53, 54]. The sections used for counting TH- or IBA1-positive cells covered the entire SN from the rostral tip of the pars compacta to the caudal end of the pars reticulate (anteroposterior, −2.06 to −4.16 mm from bregma). The counting frame was placed randomly on the first counting area and moved systematically over all counting areas until the entire delineated area was sampled. The total number of stained cells was estimated according to the optical fractionator equation [55].”

Round 2

Reviewer 1 Report

Authors have implemented most of my comments, however in the sentence:

"More recently, there are some studies reporting the link between microglia and tau in atypical parkinsonian syndromes, such as progressive supranuclear 362 palsy (PSP)" authors refer more adequately to:

 Microglial Activation and Inflammation as a Factor in the Pathogenesis of Progressive Supranuclear Palsy (PSP). Front Neurosci. 2020 Sep 2;14:893. doi: 10.3389/fnins.2020.00893. PMID: 32982676; PMCID: PMC7492584.

Author Response

Thank you so much for thoughtful reviewer. In response to reviewer's suggestion, we have changed cited reference accordingly (Page 10, Line 362). 

Reviewer 2 Report

All of my major concerns have been addressed. With the addition on in vivo IBA1 staining, this paper is much improved and strengthened.

Author Response

Thank you very much for your thoughtful comments and constructive suggestions, which have made a change to improve the quality for this manuscript. Thank you so much for the opportunity to revise our paper.